# Differential Effects of Gynecological and Chronological Age on Low Birth Weight and Small for Gestational Age

**DOI:** 10.3390/biomedicines13071639

**Published:** 2025-07-04

**Authors:** Reyna Sámano, Gabriela Chico-Barba, Estela Godínez-Martínez, Hugo Martínez-Rojano, Ashley Díaz-Medina, María Hernández-Trejo, Pablo César Navarro-Vargas, María Eugenia Flores-Quijano, María Eugenia Mendoza-Flores, Valeria Sujey Luna-Espinosa

**Affiliations:** 1Coordinación de Nutrición y Bioprogramación, Instituto Nacional de Perinatología, Secretaría de Salud Montes Urales 800, Lomas de Virreyes, Alcaldía Miguel Hidalgo, Mexico City 11000, Mexico; gabyc3@gmail.com (G.C.-B.); eygodinez@hotmail.com (E.G.-M.); maria.h.trejo72@gmail.com (M.H.-T.); pablovargas1998@gmail.com (P.C.N.-V.); maru_fq@yahoo.com (M.E.F.-Q.); tina14mx@yahoo.com (M.E.M.-F.); 2Sección de Posgrado e Investigación de la Escuela Superior de Medicina del Instituto Politécnico Nacional, Plan de San Luis y Díaz Mirón s/n, Colonia Casco de Santo Tomas, Alcaldía Miguel Hidalgo, Mexico City 11340, Mexico; 3Programa de Maestría en Ciencias de la Salud, Escuela Superior de Medicina del Instituto Politécnico Nacional, Plan de San Luis y Díaz Mirón s/n, Colonia Casco de Santo Tomas, Alcaldía Miguel Hidalgo, México City 11340, Mexico; d.ash_a@ymail.com; 4Facultad de Nutrición, Universidad Autónoma del Estado de Morelos, Cuernavaca 62350, Mexico; 5Facultad de Estudios Superiores Zaragoza, Nutriología, Universidad Nacional Autónoma de México (UNAM) Batalla 5 de mayo S/N, Ejército de Oriente Zona Peñón, Iztapalapa, Mexico City 09230, Mexico; valeriasujeylunaespinosa@gmail.com

**Keywords:** gynecological age, chronological age, menarche, adolescent pregnancy, birth weight, low birth weight, small for gestational age

## Abstract

**Background:** Adolescents with a chronological age of less than 15 years or a gynecological age of less than 2 years may have a higher probability of complications because they are more likely to conceive within 1 to 2 years of menarche and, therefore, are still growing and maturing. This could impair their ability to adapt to the physiological demands of pregnancy. **Objective:** To evaluate the relationship between chronological age and gynecological age with low birth weight and small for gestational age among adolescent mothers in Mexico City. **Methods:** A retrospective cohort design of adolescent mother–child dyads was conducted. The study followed 1242 adolescents under 19 years of age and their children, collecting data on physical, socioeconomic, and clinical characteristics, including hemoglobin levels. Low birth weight was assessed using the Intergrowth-21st project standards and categorized as above or below 2500 g. The mothers were grouped by chronological age (<15 years and ≥15 years) and gynecological age (<3 years and ≥3 years). Adjusted odds ratios were calculated using binary logistic regression models. The outcome variables were low birth weight and small for gestational age. The independent variables included gynecological age, chronological age, age at menarche, hemoglobin concentration, and gestational weight gain, among others. All independent variables were converted to dummy variables for analysis. Calculations were adjusted for the following variables: marital status, maternal education, occupation, educational lag, family structure, socioeconomic level, pre-pregnancy body mass index, and initiation of prenatal care. **Results:** The average age of the participants was 15.7 ± 1 years. The frequency of small for gestational age and low birth weight was 20% and 15.3%, respectively. Factors associated with small for gestational age included gynecological age < 3 years [aOR = 2.462, CI 95%; 1.081–5.605 (*p* = 0.032)], hemoglobin < 11.5 g/dL [aOR = 2.164, CI 95%; 1.08–605 (*p* = 0.019)], insufficient gestational weight gain [aOR = 1.858, CI 95%; 1.059–3.260 (*p* = 0.031)], preterm birth [aOR = 1.689, CI 95%; 1.133–2.519 *p* = 0.01], and living more than 50 km from the care center [aOR = 2.256, CI 95%; 1.263–4.031 (*p* = 0.006)]. An early age of menarche [aOR = 0.367, CI 95%; 0.182–0.744 (*p* = 0.005)] showed a protective effect against small for gestational age. Factors associated with low birth weight included gynecological age < 3 years [aOR = 3.799, CI 95%; 1.458–9.725 (*p* = 0.006)], maternal age < 15 years [aOR = 5.740, CI 95%; 1.343–26.369 (*p* = 0.019)], preterm birth [aOR = 54.401, CI 95%; 33.887–87.335, *p* = 0.001], living more than 50 km from the care center [aOR = 1.930, CI 95%; 1.053–3.536 (*p* = 0.033)], and early age of menarche [aOR = 0.382, CI 95%; 0.173–0.841 (*p* = 0.017), which demonstrated a protective effect, respectively. **Conclusions:** The study concludes that biological immaturity, particularly early gynecological age, significantly contributes to adverse birth outcomes during adolescent pregnancies. Interestingly, early menarche appeared to have a protective effect, whereas chronological age was not a significant predictor of small for gestational age. Chronological age has an even greater impact: women younger than 15 years are 5.7 times more likely to have low birth weight infants. However, chronological age did not increase the likelihood of having an SGA newborn.

## 1. Introduction

Adolescent pregnancy is a significant global public health issue, particularly in developing countries like Mexico, where over 21 million adolescents give birth each year, with 90% of these cases arising in such nations [1,2]. The consequences of adolescent pregnancy are severe clinical risks, including low birth weight (LBW), premature birth, intrauterine growth restriction (IUGR), and neonatal mortality [3]. Although there is a global trend of increasing age for first childbirth in developed nations, high rates of adolescent pregnancies in low-resource settings remain concerning and continue to be a leading cause of death among adolescents in low- and middle-income countries [4,5]. Some research indicates that younger maternal age may not always correlate with higher adverse perinatal outcomes [6].

The outcomes of adolescent pregnancies are affected by biological maturity and sociodemographic factors [7]. While menarche is a key milestone in reproductive development, many studies rely solely on chronological age, failing to consider biological maturity. The concept of “gynecological age (GA)” is defined as the number of years since a woman’s first menstrual period (menarche), essentially the time since the onset of reproductive capacity. It provides a more accurate measure, since the age of menarche varies among individuals [7]. Adolescent mothers may have lower height, weight, and overall maturity compared to adult women, which is compounded by the increased energy and nutrient demands of adolescence, along with socioeconomic and behavioral factors such as poverty, low education, lack of social support, and substance use [8,9,10]. These factors contribute to a higher risk of complications like premature birth, LBW, fetal abnormalities, and fetal death [11,12]. Research by Gibbs et al. [13] indicates that low GA or young maternal age at conception or delivery is predictive of LBW, linked to biological immaturity, based on the following arguments: Competition for nutrients between adolescent mothers and their fetuses can lead to adverse outcomes in the infant. Growing pregnant adolescents may prioritize their own nutritional needs over those of the fetus due to hormonal changes or insufficient intake. This competition can result in placental problems, glycine deficiency, premature birth, intrauterine growth restriction, and micronutrient deficiency, negatively affecting fetal development, especially in areas with food insecurity [14]. This study aims to explore the relationship between gynecological and chronological age with LBW and small for gestational age (SGA) among adolescent mothers in Mexico City.

## 2. Materials and Methods

### 2.1. Study Design

The study is a retrospective cohort analysis conducted at the Instituto Nacional de Perinatología (INPer) in Mexico City from January 2018 to December 2024. INPer serves as a national referral center for pregnant adolescents in Mexico City and nearby regions.

A total of 1242 adolescent mother–child pairs were included, selected through non-random sampling based on specific criteria. In the waiting room of the Obstetrics Department, all pregnant adolescents starting prenatal care who met the following inclusion criteria were invited to participate in the study: under 20 years of age; without a clinical diagnosis of chronic, autoimmune, or metabolic diseases; primiparous; with a single pregnancy; and without special diets (vegetarian, vegan, or ketogenic). If they agreed, both their parents or legal guardians and the adolescents signed the informed consent form. Subsequently, visits were scheduled to measure height and weight and administer all instruments. Given our focus on age, they were categorized by chronological age (10–15 years and 16–19 years) and gynecological age (less than 3 years and 3 or more years). Exclusion criteria included twin pregnancies, congenital malformations, underlying maternal illnesses, maternal disabilities, stillbirths, mothers not originally from Mexico, and cases of sexual abuse. Sociodemographic variables were obtained through direct questions, and the final gestational weight gain and height were measured.

### 2.2. Procedures and Data Collection

#### 2.2.1. Maternal Data

All sociodemographic information was obtained through a direct questionnaire and recorded on a registration form. Occupation was recorded as domestic worker, work outside the home, or student. Regarding educational level, information was obtained in years and grades, which allowed for the identification of educational attainment. Educational lag was defined as an adolescent being two or more grades below the expected level for her age.

Marital status was defined according to the United Nations Children’s Fund (UNICEF) criteria for child marriage, it was adjusted to combine “married” and “cohabiting” with “living with the father of her baby,” according to Mexican law. The other option was “single” when the adolescent lived without a partner [15,16].

Through a review of social work records, the type of parental family structure was determined, categorized as nuclear (cohabiting with parents and siblings), extended/composite (cohabiting with parents, siblings, grandparents, aunts, uncles, nephews, stepfather, stepmother, etc.), or single parent (cohabiting with mother and siblings, or father and siblings).

Socioeconomic status was assessed by administering the socioeconomic questionnaire of the Mexican Association of Research and Public Opinion Agencies (AMAI Rule 8 × 7) [17]. Finally, based on the address, the distance between the address and the INPer was calculated; this information was classified as urban (less than 50 km) or suburban (more than 50 km).

Chronological age was obtained from the adolescent’s birth certificate; the date of birth was used to identify whether the adolescent had early motherhood, defined as having become a mother before the age of 15 [18]. Gynecological age was calculated based on the difference between chronological age and age at menarche. A low gynecological age was defined as a pregnancy occurring within three years of menarche, according to the pelvic ripening calendar [19].

Reproductive and prenatal information: Reproductive and prenatal information was collected from the medical records. Information included age at menarche (considered early if it was less than or equal to 11 years) and gynecological age at conception [19]. Regarding prenatal care, the number of visits was classified by trimester: first trimester (≤14 weeks), second trimester (14–28 weeks), and third trimester (>28 weeks). The total number of prenatal care visits and the gestational week of each visit were recorded. Following WHO recommendations (at least eight visits to reduce perinatal mortality and improve care), the number of visits was classified as adequate (≥8) or inadequate (<8) [20].

The participants reported their pre-pregnancy weight and were measured for pre-delivery weight using precise digital scales and stadiometers (TANITA, Tokyo, Japan, model BWB-800, accuracy 0.10 kg), while height was measured using a metal stadiometer (SECA brand, Hamburg, Germany, model 208, with a precision of 0.1 cm. This information was used to compute the pre-pregnancy BMI by dividing the weight (in kg) by the square of the height (in m^2^). The BMI was then categorized into percentiles via the WHO’s AnthroPlus^®^ 1.0.4. Software (WHO AnthroPlus 1.0.4.): underweight (<3rd percentile), normal weight (3rd-85th percentile), overweight (86th-97th percentile), and obesity (>97th percentile) [21].

The total GWG was calculated by subtracting the pre-pregnancy weight from the weight measured shortly before delivery. For premature deliveries, data were obtained from clinical records. Finally, the recommended gestational weight gain for the participants were determined according to the Institute of Medicine (IOM) guidelines [22] using the following equation:*Recommended GWP (kg) at last observed weight measurement = recommended GWG for the first trimester (kg) + ((GA − 13.86) × (recommended weekly GWG in the second and third trimesters))*.

The values used in the equation were as follows: the recommended GWG for the first trimester was 2 kg (for underweight and normal pregestational BMI), 1 kg (for overweight), and 0.50 kg (for obesity). For the last two trimesters, the recommended weekly GWG was 0.51 kg (for underweight), 0.42 kg (for normal weight), 0.28 kg (for overweight), and 0.22 kg (for obesity) [22]. Gestational age was determined by counting from the first day of the pregnant adolescent’s last menstrual period (LMP) to the date of evaluation or by ultrasound. A normal pregnancy typically lasts between 38 and 42 weeks after the LMP. Using the total GWG and the recommended GWG (kg) at the time of the last weight measurement, the percentage of recommended GWG achieved was calculated using the following equation:% GWG Adequacy = (Observed GWG ÷ Recommended GWG) × 100

Finally, the percentage of recommended GWG achieved was categorized as insufficient (<90%), adequate (90–125%), and excessive (>125%) [23]. Both excessive and insufficient GWG were considered inadequate.

The pregnant adolescents first attended INPer between 20 and 26 weeks of gestational age, initiating prenatal care, which included the performance of paraclinical examinations. Among these, a complete blood count was performed to determine the hemoglobin concentration. After 8 h of fasting, a 3-milliliter whole blood sample was obtained in a Vacutainer tube containing ethylenediaminetetraacetic acid (EDTA-K2) as an anticoagulant. Hemoglobin concentrations were determined using a Beckman Coulter device, and the results were expressed in g/dL. Maternal anemia was defined as a hemoglobin level < 11.5 g/dL, according to WHO criteria [24].

#### 2.2.2. Birth Information

Delivery methods were noted. Gestational age was determined by counting from the first day of the adolescents’ last menstrual period and/or the ultrasound findings, and it was corroborated by a neonatologist through the evaluation of the newborn using the Capurro method. Births were categorized as preterm (<37 weeks), term (37–42 weeks), or post-term (>42 weeks).

#### 2.2.3. Newborn Data

Information was collected on the newborn’s sex and gestational age. Birth weight and length measurements were taken within one hour of delivery using precision equipment (measured with SECA 374, model “Baby and Mommy”; accuracy 0.1 g, and stadiometer SECA 416; accuracy 0.1 cm). Low birth weight was defined as less than 2500 g [14,25]. Birth weight classification followed the Intergrowth criteria, defining categories as SGA (<10th percentile), AGA (10th–90th percentile), and LGA (>90th percentile) [26]. All measurements adhered to the protocol of Lohman et al. [27].

### 2.3. Ethical Considerations

The study adhered to the Declaration of Helsinki and was approved by the Research, Ethics, and Biosafety Committees of the National Institute of Perinatology of Mexico. Informed consent was obtained from the parents or guardians, who signed the consent forms authorized by the committees. The adolescents signed their corresponding assent form. Review of the newborns’ medical records was also approved, and all data collection was conducted confidentially.

### 2.4. Statistical Analysis

The statistical analysis consisted of comparing the frequencies of categorical variables using the chi-squared test. The normality of the distribution of continuous variables was assessed using the Kolmogorov–Smirnov test. Descriptive statistics included measures of central tendency and dispersion, selected according to the distribution of the data. Bivariate correlations were calculated. Differences between groups were evaluated using the Mann–Whitney U test. Adjusted odds ratios (OR) were calculated using binary logistic regression models, with the outcomes being LBW and SGA. Independent variables included gynecological age < 3 years, chronological age < 15 years, age at menarche < 11 years, hemoglobin concentration < 11.5 g/dL, insufficient gestational weight gain, prematurity, initiation of prenatal care in the second or third trimester, suburban or rural residence, and height < 150 cm, all represented as dummy variables. Adjustments were made for marital status, maternal education, occupation, educational delay, family structure, socioeconomic status, pre-gestational body mass index, and initiation of prenatal care. A significance level of *p* < 0.05 was used. Analyses were performed with SPSS version 23.

## 3. Results

This study included 1242 adolescent mothers and their children between January 2018 and December 2024. The average age of the adolescents was 15.7 years (SD ± 1.0 year); 95% were between 13.7 and 17.7 years old. The average age of menarche in the adolescent mothers was 11.5 ± 1.0 years. The average gynecological age was 4.1 ± 1.0 years. Prenatal care initiation occurred in the first trimester for 1.5% of the participants, in the second trimester for 57.5%, and in the third trimester for 41%. A correlation of rho = 0.736 (*p* < 0.001) was found between birth weight (in grams) and birth weight percentile (based on Intergrowth standards).

Table 1 shows that the age of menarche was higher in adolescents older than 15 years (*p* = 0.001). The average gynecological age was 1.5 years lower among those under 15 years of age (*p* = 0.001). Birth weight was higher in adolescents older than 15 years (*p* = 0.015). The frequency of excessive gestational weight gain differed between adolescents aged 15 years or less and those older (42.5% vs. 32.6%, *p* = 0.018). Gestational age also differed significantly between the age groups (*p* < 0.001).

Regarding socioeconomic variables such as schooling, adolescents aged 15 years or older had higher levels of schooling than younger adolescents. However, no significant differences were observed in educational lag between the two groups. Finally, a higher percentage of adolescents aged 15 years or older cohabited with their partner.

Maternal characteristics according to gynecological age (<3 years vs. ≥3 years) can be seen in Table 2. It shows that adolescents with a gynecological age of less than three years had an average age of menarche of 12.6 years, compared to adolescents with a gynecological age greater than or equal to three years, who had an average age of menarche of 11.3 years (*p* = 0.001). In addition, an association between a higher percentage of adequacy in gestational weight gain in adolescents with gynecological age less than 3 years, compared to adolescents with gynecological age greater than or equal to 3 years, as well as an association between a lower hemoglobin concentration and a lower gestational age among those with lower gynecological age, compared to those with a gynecological age greater than or equal to 3 years, is highlighted. Among the sociodemographic variables, schooling, family structure, and marital status showed an association with differences in their frequencies according to gynecological age (see Table 2).

Table 3 shows that approximately 50% of mothers with SGA newborns had insufficient GWG (*p* = 0.020). After adjustment in the regression models, this persisted as an independent variable associated with a higher probability of having a newborn small for gestational age. Regarding hemoglobin concentration, among women with SGA newborns, 34% had anemia and 61% did not. Among babies with normal weight, 10% had mothers with anemia and 90% did not. Furthermore, a higher frequency of SGA newborns was observed in mothers with premature pregnancies, of low socioeconomic status, and residing in suburban or rural communities.

The average weight of newborns from adolescent mothers under 14 years of age was 2838 ± 362 g. The group with the highest average weight was that of adolescent mothers aged 16 years, with 2928 ± 535 g, observing a tendency that the older the mother, the higher the weight of the newborn (see Figure 1).

One hundred and ninety pregnant adolescents, representing 15.3% of the sample studied, had newborns with low weight (<2500 g). When using Intergrowth criteria to evaluate intrauterine growth restriction, it was found that 254 pregnant adolescents, corresponding to 20.45% of the sample, had an SGA newborn. See Table 3 and Table 4.

Mothers with a height of less than 150 cm presented a higher proportion of newborns with low birth weight. Likewise, a higher prevalence of hemoglobin values of less than 11.5 g/dL stood out in mothers whose newborns presented low birth weight, as well as a higher incidence of preterm pregnancies in this same group. (See Table 4).

Table 5 shows that the following variables were associated with a significantly increased probability of an SGA birth: gynecological age less than 3 years, hemoglobin concentration below 11.5 g/dL during the third trimester, preterm birth, insufficient gestational weight gain (GWG), and residence in a suburban or rural area more than 50 km from the hospital used for pregnancy control and delivery care. Menarche before age 11, in contrast, was a protective factor.

For LBW, factors significantly increasing the probability included gynecological age less than 3 years, chronological age of 15 years or less, preterm birth, and living in a suburban or rural area more than 50 km from INPer. As in the model for SGA, menarche before age 11 was associated with a 62% reduction in the probability of LBW. The remaining variables were not significant.

## 4. Discussion

The results of the present study support our hypothesis regarding the effect of biological immaturity on the presence of LBW and intrauterine growth restriction (IUGR). The main variables associated with LBW and being SGA are biological immaturity, represented by a gynecological age (time elapsed from menarche to the time of pregnancy) of less than 3 years and a chronological age equal to or less than 15 years.

Adolescent mothers with a low gynecological age are 3.8 times more likely to have low birth weight babies than those with a gynecological age of 3 years or older. However, this effect is smaller than that associated with chronological age. The increased probability of low birth weight in adolescents under 15 years of age is likely related to the higher probability of preterm birth in this group. Moreover, a chronological age under 15 years was associated with low birth weight, but not SGA, which is also likely influenced by the greater likelihood of preterm birth. Adolescent mothers with a gynecological age under 3 years have a roughly 2.5 times greater probability of having an SGA newborn.

### 4.1. Low Birth Weight and Small for Gestational Age

In the study, the prevalence of low birth weight (LBW) and newborns small for gestational age (SGA) in adolescent mothers was 15.3% and 20%, respectively, figures that exceed the upper limit of 15% reported for Mexico and the global average of 14.7% in 2020 [28]. They are higher than the prevalence in Canada, where 7.4% was reported [29,30], but lower than that of Malaysia, with 19.3% [31]. Worldwide, in 2020, there were approximately 19.8 million LBW births, predominantly in South Asia and sub-Saharan Africa, which accounted for 44.5% and 27.1% of these cases, respectively [32].

LBW and being SGA indicate delays in fetal growth and are linked to the duration of pregnancy and birth outcomes, such as an increase in preterm birth and cesarean delivery [33]. Both increase the probability of complications throughout life, such as developmental delays, increased susceptibility to infections, learning problems, and behavioral disorders [34]. LBW is a significant cause of neonatal mortality and can have negative effects on long-term health, including growth and neurological development problems, as well as an increased probability of chronic diseases such as type 2 diabetes and cardiovascular disorders [35,36]. In addition, the consequences of LBW and SGA negatively affect families and society. The WHO has proposed reducing the prevalence of LBW by 30% globally between 2012 and 2025, but Mexico faces challenges in reaching this goal, making it crucial to better understand the factors that contribute to this problem, especially in adolescent mothers [37].

### 4.2. Age of Menarche

The average age at menarche differed significantly, being higher in adolescent mothers with a gynecological age greater than 3 years compared to those with a gynecological age less than 3 years. This difference is significant and comparable to the 12 years reported by Hinojosa-Gonzalez and Robellada-Zárate in adolescents from northeastern Mexico [38,39]. Marván et al. found an average age of 11.4 years in students from Mexico City [40], which is one year younger than that reported in studies from Colombia (12.6 years) [41], Korea (12.7 years) [42], and the United States (12.8 years in white girls and 12.2 years in black girls) [43]. The results indicate that the age of menarche is relevant, as early menarche is associated with a higher probability of low birth weight if the adolescent becomes pregnant within the following three years [44]. Therefore, a gynecological age greater than 3 years and a chronological age greater than 16 years are recommended to reduce this probability.

Contrary to expectations, menarche before age 11 appears to be a protective factor against low birth weight (LBW) and small for gestational age (SGA). This may be explained by the fact that the average chronological age of adolescents at pregnancy in our study was 15.7 years. In this context, while the group of adolescents under 15 years of age experienced menarche, on average, at 11.3 years, they still had a lower probability of reaching a gynecological age greater than 3 years compared to adolescents over 15. The time elapsed between menarche and pregnancy allows most adolescents to achieve a certain level of biological maturity before becoming pregnant [45,46]. Furthermore, the early onset of menarche does not necessarily correlate with earlier sexual activity or an earlier age at first pregnancy, as demonstrated by Marino JL in a group of 541 Australian adolescents [47].

Furthermore, in a Mexico City hospital, the mean age at menarche was 11.7 ± 1.4 years, and the age at first sexual intercourse was 15.0 ± 1.3 years in a group of 608 adolescents [48]. Hinojosa-González et al. [38] reported that the mean age at menarche was 12 years, and the age at first sexual intercourse was 16 years in a group of 814 adolescents from northeastern Mexico. These results, which show that the adolescents were gynecologically older than 3 years, support the concept of biological maturity.

Therefore, it is essential to study the impact of religiosity and other sociocultural factors that possibly delay the onset of sexual intercourse in adolescents with early menarche. This has not been fully explored, and further studies are needed to clarify the impact of age at coitarche on adolescent pregnancy and newborn weight.

In Mexico, a significant decrease in the age of onset of menarche has been observed over the decades, with a reduction of 1.35 months per decade [2,40]. This change has various implications for public health and gynecological medical care. However, considering data from the last 50 years, these trends no longer appear significant in developed countries [49]. However, recent studies show that Mexican adolescents are experiencing menarche at an earlier age [40]. Early onset of menstruation has been associated with a higher risk of health problems, including breast cancer, cardiovascular diseases, and reduced life expectancy [42]. Nutritional status and genetic factors could be significant, demonstrating the influence of body fat in childhood and adolescence on the timing of sexual maturation and the involvement of a specific genetic variant that contributes to early menarche in obese girls [42].

### 4.3. Gynecological Age

Our study demonstrated that the gynecological age of an adolescent mother significantly influences the baby’s birth weight and the likelihood of the baby being SGA. Specifically, adolescents with less than three years of gynecological age are almost four times more likely to give birth to babies with LBW compared to adolescent mothers with a gynecological age of three years or more. Our results are consistent with those reported in a systematic review by Gibbs CM et al. [13], as well as with the study conducted by Masyitah S and Kusharisupeni [50].

Furthermore, the concept of gynecological age reflects the biological maturation of pregnant adolescents and allows for numerical documentation that is used to increase the accuracy of observations and provide standardization [19,51]. Low gynecological age has a direct negative effect on fetal development due to the high maternal nutritional requirements and is one of the main causes of LBW [52]. Two general characteristics of biological immaturity could play a role in increasing the probability of adverse outcomes: a young gynecological age [53] and the pregnancy of an adolescent before her own growth has ceased [54,55]. Immaturity of the uterine or cervical blood supply may predispose adolescent mothers to subclinical infections, an increase in prostaglandin production, and, consequently, a higher incidence of premature delivery. Adolescent mothers who continue to grow during pregnancy may compete with the developing fetus for nutrients, to the detriment of the fetus. This assumption is supported by evidence indicating that weight gain during pregnancy may be more critical for adolescent mothers than for adult mothers [56,57]. Consequently, the intrinsic probability of adverse pregnancy outcomes among adolescents is likely partly attributable to early gynecological age or inadequate weight gain. Thus, low gynecological age has been associated with higher rates of adverse reproductive outcomes in adolescents and may be more closely related to biological outcomes than chronological age [58]. Furthermore, our study demonstrates that a gynecological age of less than 3 years in adolescent pregnancies significantly increases the likelihood of low birth weight and intrauterine growth restriction.

### 4.4. Early Motherhood in Adolescence

Since 2002, Phipps, MG, et al. have analyzed data from a study of 768,029 women aged 12–23 years from the 1995 US birth cohort (births occurring within the United States) [18]. Their findings suggest that the probability of infant mortality increases significantly when the mother is between 13 and 14 years old. However, when considering factors such as low birth weight, prematurity, and infant death, the optimal cutoff point is between 15 and 16 years old. Younger adolescents (mothers aged 15 years and younger) have higher rates of very low birth weight babies, prematurity, and infant deaths compared with mothers aged 16–19 years [18]. These data are consistent with our results, which reveal that pregnant adolescents under 15 years of age have a 5.7-fold increased probability of having a low-birth-weight newborn, but not with intrauterine growth restriction. Furthermore, this group is associated with preterm births and a higher frequency of maternal anemia compared with older adolescents, which is also consistent with a study conducted in rural Nepal [59].

Likewise, younger adolescent mothers have higher rates of delayed antenatal care initiation and insufficient weight gain compared with mothers aged 16–19 years [60,61]. However, these results contrast with our findings, which showed no association between the initiation of antenatal care, as it was late in both groups (after 24 weeks of gestation), nor was there a difference in insufficient gestational weight gain. There are multiple possible explanations for the low use of antenatal care and limited weight gain during pregnancy [62]. Both factors have been associated with increased exposure to physical violence before and during pregnancy [63]. Therefore, it is important to investigate the relevance of these factors in the early childbearing age group of adolescents.

The factors associated with childbirth vary between younger and older adolescent mothers. Although younger adolescents have lower cesarean section rates, no significant differences were found compared to mothers aged 16–19 years and adults [12]. This could be because the National Institute of Perinatal Health (INPer), as a teaching hospital, classifies adolescent pregnancy as high-risk due to factors such as short stature and a greater likelihood of cephalopelvic disproportion. Adolescents also have a higher incidence of maternal anemia, possibly related to poor nutritional status [64,65]. While the theory of biological immaturity exists as a risk factor [66], it is argued that it is not the only cause of low birth weight, as it is also influenced by chronological and gynecological age as well as geographic location (which might affect access to adequate nutrition and healthcare).

### 4.5. Anemia

Our study identified that newborns of adolescent mothers with anemia during pregnancy had a higher probability of being SGA. Surprisingly, although LBW was observed in anemic adolescent mothers, we found no significant association between a hemoglobin concentration below 11.5 g/dL and LBW. Our results differ from those of some systematic reviews and meta-analyses conducted in other countries [67,68,69,70,71,72,73]. However, these findings align with other meta-analyses that have shown no effect of maternal anemia during pregnancy on LBW [74] or no significant association between maternal hemoglobin concentrations below 11.5 g/dL and LBW [75].

These differences could be due to variations in the sociodemographic characteristics of the study participants [75]. They could also be influenced by the fact that, in our study, anemia was diagnosed, on average, at week 24 of gestation, during the adolescents’ first prenatal visit. Therefore, it is possible that anemia did not significantly affect the newborn’s birth weight; alternatively, anemia was present earlier but was not detected until it was measured at the time the adolescent first attended prenatal care. It should be noted that most of the reviewed studies are based on adult women.

The explanation for LBW and having an SGA newborn lies in the fact that anemia reduces the oxygen supply to the fetus by decreasing hemoglobin concentrations, impairing placental function, and altering angiogenesis [76]. This oxygen deficit can restrict fetal growth, leading to adverse pregnancy outcomes such as premature delivery and LBW [77]. In addition, inadequate nutrient intake during pregnancy, which is often observed in anemic pregnant adolescents, further increases the risk by depriving the fetus of essential nutrients for growth [78].

### 4.6. Sociodemographic Factors

In an initial unadjusted analysis, a significant association was found between socioeconomic factors and LBW or SGA. However, after adjusting for other factors, only suburban or rural residence maintained a significant association with LBW or SGA. Living in a suburban or rural residence implies less access to medical services, and adolescents residing in these areas often come from a low or very low socioeconomic level, which likely limits their nutrition. All of this affects fetal growth and gestational age, resulting in more frequent premature births.

The difficulty in demonstrating an independent association of low birth weight or being small for gestational age with sociodemographic characteristics such as lower educational level and academic aspirations, scarcity of economic resources, the influence of family relationships (and other relationships important for adolescents), urbanization and anonymity in large cities, excess free time, and both parents working is probably due to the fact that we studied a homogeneous sample of pregnant adolescents who, for the most part, come from a very similar sociodemographic background, which does not allow us to completely isolate these effects. Furthermore, given that the study population comes from a national referral hospital, it may be biased towards a higher biological risk. This could amplify the observed effects of adolescent age and, in turn, minimize the influence of socioeconomic factors, which are more common in the general population.

On the other hand, it should be noted that adolescent mothers face greater challenges than adult mothers, including higher rates of poverty, stress, mental health problems, substance use, and an increased risk of post-traumatic stress disorder [79]. These socioeconomic disadvantages are associated with lower educational levels [80].

Numerous investigations indicate that poverty and socioeconomic and demographic disadvantage are relevant factors in the correlation of obstetric complications and adverse birth outcomes in adolescent pregnancies [81]. However, in recent years, it has also been shown that biological immaturity has been associated with low birth weight or being small for gestational age [13,66], as demonstrated in our study.

While it is true that many pregnant adolescents face unfavorable environments characterized by poverty, lack of access to education and adequate medical care, and limited support networks, evidence suggests that biological immaturity also contributes significantly to the probability of low birth weight and being small for gestational age. In conclusion, the etiology of low birth weight or being small for gestational age is multifactorial and is related to sociodemographic, nutritional, biological, and environmental aspects.

### 4.7. Strengths and Limitations

Among the main strengths of this study are the large sample size, the availability of detailed information, and the inclusion of all subgroups of age, pre-pregnancy BMI, gestational weight gain, gestational age, and newborn weight. However, it is important to consider some limitations. The generalizability of our prevalence rates and odds ratios may be limited by several factors. These include the recruitment of adolescent mothers from a single medical center in Mexico City, the use of a nonprobability sample of consecutive cases meeting the selection criteria, and the diversity in nutritional status, socioeconomic levels, and the initiation and quality of care among adolescents. To demonstrate the generalizability of our results, it is critical that future multicenter studies include adolescents from different regions of Mexico using probability sampling. However, the National Institute of Perinatology is a hospital center that serves a considerable number of pregnant adolescents from both Mexico City and neighboring states. Another limitation was the use of the recall method to assess the age at menarche; however, since the adolescents were assessed at ages close to the event and the interview was conducted by a health professional, the risk of recall bias was minimized, as demonstrated by Dorn LD et al. [82] and Mao Y et al. [83].

### 4.8. Implications

Our study contributes significantly to the understanding of the determinants of low birth weight in children of adolescent mothers, providing relevant information for clinical practice and individualized care. The results allow for the identification of pregnant adolescents with a higher probability of having a child with low birth weight, which facilitates the implementation of early preventive and therapeutic measures. For example, knowledge of the influence of gynecological age and maternal anemia can guide nutritional supplementation and intensive prenatal monitoring.

To optimize clinical care, future research should explore in-depth factors not yet studied, such as the influence of subclinical infectious diseases and the impact of specific nutritional deficiencies on fetal development. In addition, it is essential to investigate the effectiveness of different communication and counseling strategies to improve the adherence of pregnant adolescents to prenatal care programs and promote healthy lifestyles.

Additionally, Mexico should focus its investments on programs aimed at preventing adolescent pregnancy and low birth weight, improving care during premature delivery, and addressing intrauterine growth restrictions, which are underlying factors that contribute to low birth weight. It is important to consider that pregnancy in girls under 15 years of age, regardless of the cause, puts their physical, mental, and economic well-being, as well as that of their children, at risk. Therefore, it is crucial that public health and education policies facilitate access to information and resources for responsible sexuality, protect the rights of girls and women, and prevent violence effectively.

## 5. Conclusions

Low birth weight (LBW) and small for gestational age (SGA) in infants of adolescent mothers are related to maternal biological immaturity. Both gynecological age and chronological age are significantly associated with these outcomes. Mothers with a gynecological age of less than 3 years since menarche are 3.8 times more likely to have infants weighing less than 2500 g and 2.5 times more likely to have an SGA newborn. Chronological age has an even greater impact: women younger than 15 years are 5.7 times more likely to have low birth weight infants. However, chronological age did not increase the likelihood of having an SGA newborn. Contrary to expectations, menarche before the age of 11 appears to be a protective factor for both low birth weight and SGA.

The study highlights the importance of managing maternal anemia in pregnant adolescents, as this condition significantly increases the likelihood of having an SGA newborn. To prevent adverse pregnancy outcomes in low- and middle-income countries like Mexico, it is essential to

Prioritize the treatment of maternal anemia;Reduce the number of adolescent pregnancies;Provide adequate prenatal care to young pregnant women.

During prenatal care, pregnant adolescents should be informed about potential complications and recommended to give birth in experienced centers to avoid adverse perinatal outcomes.

## Figures and Tables

**Figure 1 biomedicines-13-01639-f001:**
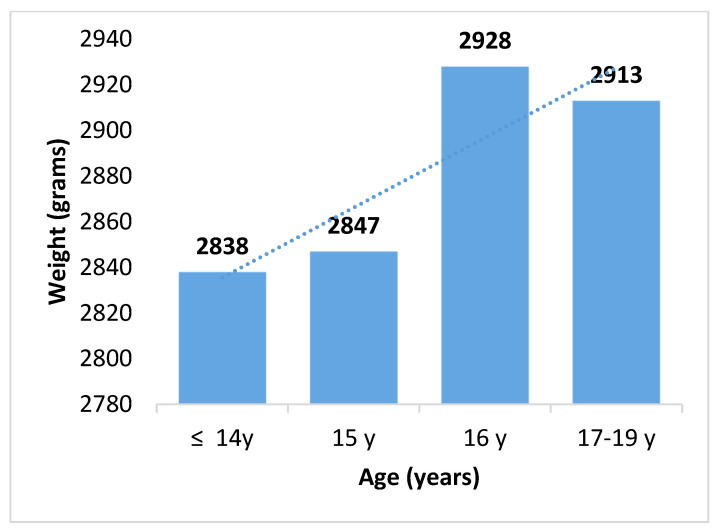
Birth weight (grams) according to the mother’s age (*p* = 0.026 for the correlation).

**Table 1 biomedicines-13-01639-t001:** General characteristics of pregnant adolescents according to chronological age.

Variable	Chronological Age, *n* (%)		*p*-Value *
	<15 y, *n* = 491 (40)	≥15 y, *n* = 751 (60)	
Menarche age (y) ^a^	11.3 ± 1	11.7 ± 1	0.001
Gynecological age (y) ^a^	3.1 ± 1	4.8 ± 1	0.001
Pre-pregnancy weight (kg) ^a^	51 (46–58)	51 (47–58)	0.359
Height (cm) ^a^	155.5 ± 5	155.7 ± 5	0.657
Height			
Less than 150 cm	79 (16)	122 (16)	0.942
pBMI (kg/m^2^) ^a^	21.5 ± 3	21.7 ± 3	0.262
Low weight	98 (20)	113 (15)	0.039
Normal weight	332 (68)	530 (70)	
Overweight	51(10)	89 (12)	
Obesity	10 (2)	19 (3)	
Final gestational weight (kg) ^b^	64 (58–72)	63 (58–71)	0.111
Gestational weight gain (kg) ^b^	13 (9–16)	12 (8–16)	0.551
Adequacy of GWG (%) ^b^	110 (79–158)	98 (71–140)	0.018
Hemoglobin (g/dL) ^a^	12.4 ± 1	12.6 ± 1	0.009
Start of prenatal care (weeks)	24 (20–30)	25 (20–29)	0.097
1st trimester	10 (2)	9 (1)	0.505
2nd trimester	368 (75)	570 (76)	
3rd trimester	113 (23)	172 (23)	
Less than 8 antenatal visit	478 (97)	741 (99)	0.073
8 and more antenatal visit	13 (3)	10 (1)	
Mode of delivery			
Cesarean section	273 (56)	385 (51)	0.075
Gestational age at delivery (weeks)	38.2 ± 2	39 ± 2	0.001
Birth weight (g)	2877 (2688–3070)	2943 (2670–3206)	0.015
Small for gestational age (<10 percentile)	92 (19)	162 (22)	0.260
Adequate for gestational age (10–90 percentile)	392 (80)	578 (77)	
Large for gestational age (>90 percentile)	7 (1)	11 (1)	
Low birth weight (≤2500 g)	74 (15)	116 (15)	0.016
Adequate birth weight (2501–3999 g)	401 (82)	583 (78)	
Macrosomic (≥4000 g)	16 (3)	52 (7)	
Sociodemographic variables			
Occupation			0.358
Homemaker/housekeeper	434 (88)	670 (89)	
Works outside the home or student	26 (14)	31 (10)	
Educational level			0.001
Elementary	277 (56)	153 (20)	
Secondary	214 (44)	560 (75)	
High school	0 (0)	38 (5)	
Educational lag	251 (51)	363 (48)	0.184
Socioeconomic level			0.163
Middle low	357 (73)	566 (75)	
Very low	134 (27)	185 (25)	
Family structure			0.320
Nuclear	263 (56)	389 (55)	
Extended/blended	205 (44)	323 (45)	
Marital status			0.001
Cohabiting	170 (35)	335 (45)	
Single	321 (65)	416 (55)	
Residence			0.440
Urban (<50 km)	481 (64)	319 (65)	
Suburban or rural (>50 km)	270 (36)	172 (35)	

* *p*-value by Pearson’s chi-squared test. (a) Data expressed as mean ± SD, *p*-value by Student’s *t*-test; (b) data expressed as median (25th percentile-75th percentile), *p*-value by Mann–Whitney U test. GWG: gestational weight gain; pBMI: pre-pregnancy body mass index.

**Table 2 biomedicines-13-01639-t002:** General characteristics of pregnant adolescents according to gynecological age.

Variable	Gynecological Age, *n* (%)		*p*-Value *
	<3 y, *n* = 175 (14)	≥3 y, *n* = 1067 (86)	
Menarche age (y) ^a^	12.6 ± 1	11.3 ± 1	0.001
Pre-pregnancy weight (kg) ^a^	51 (46–58)	51 (47–58)	0.359
Height (cm) ^a^	155.6 ± 5	155.7 ± 5	0.657
Less than 150 cm	26 (15)	175 (16)	0.349
pBMI (kg/m^2^) ^a^	21.6 ± 3	21.6 ± 3	0.097
Low weight	36 (21)	175 (16)	0.218
Normal weight	110 (63)	752 (71)	
Overweight	25 (14	115 (11)	
Obesity	4 (2)	25 (2)	
Final gestational weight (kg)	64 (56–74)	63 (58–71)	0.579
Gestational weight gain (kg)	13 (10–16)	12 (8–16)	0.551
Adequacy of GWG (%)	110 (79–167)	103 (72–143)	0.018
Hemoglobin (g/dL) ^a^	12.4 ± 1	12.6 ± 1	0.008
Start of prenatal care (weeks) ^b^	25 (20–31)	25 (20–29)	0.538
1st trimester	4 (2)	15 (1)	0.679
2nd trimester	131 (75)	807 (76)	
3rd trimester	40 (23)	245 (23)	
Less than 8 antenatal visit	171 (98)	1048 (98)	0.411
8 and more antenatal visit	4 (2)	19 (2)	
Mode of delivery			
Cesarean- section	81 (46)	503 (47)	0.449
Gestational age at delivery (weeks) ^a^	38 ± 1	39 ± 1	0.001
Bith weight (g) ^b^	2826 (2648–3022)	2952 (2682–3214)	0.001
Small for gestational age (<10 percentile)	38 (21.5)	216 (20)	0.475
Adequate for gestational age (10–90 percentile)	136 (78)	834 (78)	
Large for gestational age (>90 percentile)	1 (0.5)	17 (2)	
Low birth weight (≤2500 g)	30 (17.2)	160 (15)	0.008
Adequate birth weight (2501–3999 g)	144 (82.3)	840 (79)	
Macrosomic (≥4000 g)	1 (0.5)	67 (6)	
Sociodemographic variables			
Occupation			0.153
Homemaker/housekeeper	160 (91)	944 (89)	
Work out at home or student	15 (9)	123 (11)	
Educational level			
Elementary	97 (55)	333 (31)	0.001
Secondary	77 (44)	697 (65)	
High school	1 (0.5)	37 (4)	
Educational lag	87 (50)	527 (49)	0.501
Economic level			
Middle low	128 (73)	795 (74)	0.382
Very low	47 (27)	272 (26)	
Family structure			
Nuclear	109 (62.3)	576 (54)	0.036
Extended/blended	66 (37.7)	491 (46)	
Marital status			
Cohabiting	56 (32)	449 (42)	0.007
Single	119 (68)	618 (58)	
Residence			0.320
Urban (<50 km)	105 (60)	693 (65)	
Suburban or rural (>50 km)	70 (40)	374 (35)	

* *p*-value by Pearson’s chi-squared test. (a) Data expressed as mean ± SD, *p*-value by Student’s *t*-test; (b) data expressed as median (25th percentile-75th percentile), *p*-value by Mann–Whitney U test. GWG: gestational weight gain; pBMI: pre-pregnancy body mass index.

**Table 3 biomedicines-13-01639-t003:** Comparison of characteristics according to fetal growth (Intergrowth classification).

Variable	Fetal Growth, *n* (%)		*p*-Value
	SGA, *n* = 254 (20)	Adequate + LGA, *n* = 988 (80)	
Maternal age			0.679
<14 y	34 (13)	152 (15)	
15 y	58 (23)	247 (25)	
16 y	100 (40)	366 (38)	
17–19 y	62 (24)	223 (22)	
Height			0.152
<150 cm	47 (18)	154 (16)	
≥150 cm	207 (82)	834 (84)	
pBMI			0.056
Low weight	52 (20)	159 (16)	
Normal weight	166 (66)	696 (70)	
Overweight	34 (13)	106 (11)	
Obesity	2 (1)	27 (3)	
Gestational weight gain			0.020
Insufficient	118 (46)	365 (37)	
Adequate	58 (23)	278 (28)	
Excessive	78 (31)	345 (35)	
Hemoglobin (g/dL)			0.001
<11.5	100 (34)	101 (10)	
≥11.5	154 (61)	887 (90)	
Gynecological age (y)			0.360
<3	38 (15)	137 (14)	
≥3	216 (85)	851 (86)	
Menarcheal age (y)			0.001
≤11	88 (35)	466 (47)	
>12	166 (65)	522 (53)	
Gestational age at birth			0.001
Term	213 (84)	878 (89)	
Preterm	41 (16)	110 (11)	
Mode of delivery			0.033
Vaginal	121 (48)	537 (54)	
Cesarean-section	133 (52)	451 (46)	
Start of prenatal care			
1st trimester	1 (0.4)	18 (2)	0.202
2nd trimester	190 (74.8)	748 (76)	
3rd trimester	63 (24.8)	222 (22)	
Less than 8 antenatal visits	251 (99)	968 (98)	0.276
8 and more antenatal visits	3 (1)	20 (2)	
Socioeconomic level			0.001
Middle low	159 (63)	764 (77)	
Very low	95 (37)	224 (23)	
Marital status			0.295
Cohabiting	99 (39)	406 (41)	
Single	155 (61)	582 (59)	
Residence			0.001
Urban (<50 km)	124 (49)	692 (70)	
Suburban or rural (>50 km)	130 (51)	296 (30)	

Data expressed as frequency (%). *p*-values from Pearson’s chi-square test. GW: Gestational week.

**Table 4 biomedicines-13-01639-t004:** Comparison of characteristics by fetal growth (<2500 g, ≥2500 g).

Variable	Birth Weight by 2500 g as Reference, *n* (%)		*p*-Value *
	LBW, *n* = 190 (15.3)	NLBW, *n* = 1052 (84.7)	
Maternal age			0.010
<14 y	26 (14)	160 (15)	
15 y	48 (25)	257 (24)	
16 y	56 (29)	410 (39)	
17–19 y	60 (32)	225 (21)	
Height			0.021
<150 cm	41 (22)	160 (15)	
≥150 cm	149 (78)	892 (85)	
pBMI			0.184
Low weight	12 (6)	37 (4)	
Normal weight	150 (79)	815 (78)	
Overweight	20 (11)	142 (14)	
Obesity	8 (4)	58 (6)	
Gestational weight gain			0.796
Insufficient	78 (41)	405 (39)	
Adequate	49 (26)	287 (27)	
Excessive	63 (33)	360 (34)	
Hemoglobin (g/dL)			0.001
<11.5	56 (30)	145 (14)	
≥11.5	134 (70)	907 (86)	
Gynecological age (y)			0.264
<3 y	30 (16)	145 (14)	
≥3 y	160 (84)	907 (86)	
Menarcheal age (y)			0.161
<11	78 (41)	476 (45)	
≥11	112 (59)	579 (55)	
Gestational age at birth			0.001
Term	72 (38)	1019 (97)	
Preterm	118 (62)	33 (3)	
Mode of delivery			0.006
Vaginal	82 (43)	576 (55)	
Cesarean section	108 (57)	476 (45)	
Start of prenatal care			
1st trimester	3 (2)	16 (2)	0.139
2nd trimester	154 (81)	784 (74)	
3rd trimester	33 (17)	252 (24)	
Less than 8 antenatal visits	187 (98)	1032 (98)	0.523
8 and more antenatal visits	3 (2)	20 (2)	
Marital status			0.139
Cohabiting	70 (37)	435 (41)	
Single	120 (63)	617 (59)	
Socioeconomic level			0.028
Middle low	130 (68)	793 (75)	
Very low	60 (32)	259 (25)	
Residence			0.002
Urban (<50 km)	93 (49)	726 (69)	
Suburban (>50 km)	97 (51)	326 (31)	

Data expressed as frequency (%). * *p*-values from Pearson’s chi-square test. GW: Gestational week; LBW: low birth weight; NLBW: no low birth weight.

**Table 5 biomedicines-13-01639-t005:** Regression models for small for gestational age (SGA) and for low birth weight (<2500 g).

Variable	aOR *	CI 95%	*p*-Value
Small for Gestational Age			
Gynecological age < 3 y	2.462	1.081–5.605	0.032
Age < 15 y	1.149	0.404–3.264	0.794
Menarcheal age < 11 y	0.367	0.182–0.744	0.005
Hemoglobin < 11.5 g/dL	2.164	1.081–5.605	0.019
Insufficient GWG	1.858	1.059–3.260	0.031
Preterm birth	1.689	1.133–2.519	0.010
Start antenatal care in 2nd or 3rd trimester	4.695	0.624–35.335	0.133
Suburban or rural residence	2.256	1.263–4.031	0.006
Height < 150 cm	0.907	0.448–1.837	0.786
**Low Birth Weight**			
Gynecological age < 3 y	3.799	1.458–9.725	0.006
Age < 15 y	5.740	1.343–26.369	0.019
Menarche age < 11 y	0.382	0.173–0.841	0.017
Hemoglobin < 11.5 g/dL	1.965	0.992–3.892	0.053
Insufficient GWG	1.633	0.902–2.956	0.105
Preterm birth	54.401	33.887–87.335	0.001
Start antenatal care in 2nd or 3rd trimester	4.221	0.325–54.776	0.271
Suburban or rural residence	1.930	1.053–3.536	0.033
Height < 150 cm	1.794	0.903–3.566	0.095

* Adjusted for socioeconomic level, occupation, pre-pregnancy BMI, cesarean section, family structure, marital status, educational attainment, and educational level.

## Data Availability

The data presented in this study are available from the corresponding author upon reasonable request.

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
