# Peer review of "Differential Effects of Gynecological and Chronological Age on Low Birth Weight and Small for Gestational Age"

_biomedicines, 2025, doi:10.3390/biomedicines13071639_

Round 1

Reviewer 1 Report

Comments and Suggestions for Authors

Biomedicines

Review of Manuscript

Biomedicines-3704218:

Low birth weight or small for gestational age in neonates of adolescent mothers:

Effect of gynecological and chronological age

General Impressions:

Adolescents’ increased risk for adverse pregnancy outcomes is a major public health concern, particularly in lower resourced countries where adolescent pregnancy remains relatively high.  Examining a spectrum of biological and socio-ecological factors that increase risk can provide important insights for promoting healthier reproductive outcomes in adolescents. The main objective of this study is to examine the relation of chronological age and gynecological age with low birth weight (LBW) and small-for-gestational-age (SGA) in a sample of adolescent mothers, adjusting for biological and socio-ecological risk factors. The main finding that gynecological age is independently associated with LBW and SGA emphasizes the importance of considering biological immaturity, not just chronological age, in studies of adolescent pregnancy and in public health and health care efforts to promote healthier outcomes.  Overall, the manuscript is well-written, the research methods seem appropriate, the literature cited is mostly within 5-10 years of publication, and the conclusions follow from the results.  This study could make an important contribution to the literature in this area. Improving clarity in the Methods will enhance the manuscript for publication.

Specific Comments:

 Abstract

  • Background: Authors could highlight the importance of considering biological maturity (ie, gynecological age) in addition to chronological age in adolescent pregnancy.
  • Objective: Authors should include SGA in the study objective.
  • Methods:
    • Authors might specify that the study uses a retrospective cohort design, and that low birth weight was assessed using Intergrowth standards. As a retrospective study, they also might want to edit the statement that they “followed” 1242 adolescents.
    • There seems to discrepancies between the list of control variables in the Abstract and those noted in Table 5 where the regression results are reported.
  • Results:
    • Authors might check whether the journal requires all of the regression-related statistics to be reported in the Abstract.
    • Given their study objective, authors also could point out that chronological age was not a significant predictor of LBW
  • Conclusion:
    • Authors might place the menarche statement after the initial conclusion about biological immaturity, since it is part of the results.
    • The importance of considering sociodemographic factors in prenatal care is a bit confusing, because these were used primarily as control variables, rather than being the focus of study.
  • Keywords: SGA is missing. It is unclear why pathogenesis is included.

 Introduction

  • Lines 77-78: A definition of gynecological age would be quite helpful to readers.
  •  
  • Lines 83-84: Authors could clarify results of Gibbs et al, i.e., which factor – low GA or young maternal age – was predictive of LBW? How was the result linked to biological immaturity in their study?

  • Lines 85-86: It is unclear what the debate is between biological and socioenvironmental factors and why that would be mentioned here, given that the study’s focus is stated in the next line: relations of gynecological and chronological age to LBW and SGA.

Materials and Methods

  • Lines 91-99: The manuscript would benefit from more details about the study design and the sampling procedures. Were all data collected through record reviews, for example, or a combination of records and interviews or surveys with the participants?  How were participants recruited and what non-random methods were used to create the sample? Authors might consider moving the “INPer” statement from the end of the section to the beginning.

  • Lines 100-155: Procedures and Data Collection section only includes information on measurement of study variables. Including details about the study procedures would enhance the manuscript. Perhaps, this can be addressed in Lines 91-99.  Defining acronyms prior to use also would be helpful to readers.

  • Line 135: Gestational age traditionally is measured based on last menstrual period and date of delivery. Authors might explain why they assessed gestational age based on time of last prenatal care visit.

  • Lines 142-144: Table 3 indicates hemoglobin was assessed in 3rd trimester; this could be clarified here where the measure is initially described. Lower hemoglobin in 3rd trimester may not be indicative of hemoglobin levels throughout the pregnancy course.

  • Lines 146-147: Definition of gestational age offered here is inconsistent with earlier definition in Line 135.

  • Lines 150-151: Gestational age is described a third way here.

  • Lines 168-172: Authors ensure that the list of independent and control variables mentioned here match those listed in the model in Table 5. It also is unclear how variables were selected for the regression models. Based on the bivariate analyses, there were other variables significantly associated with the outcomes (e.g., mode of delivery), which were not included in the models.

  • Readers may find Figure 1 somewhat confusing. It is labeled as the distribution of chronological age and gynecological age, but it depicts age at menarche and chronological age.  The overlap of the distributions between ages 12-15 also presents some challenges in understanding the figure. Authors might consider whether this figure is necessary.

  • Lines 184-185: The two outcome variables are missing from Table 1 (and Table 2). LBW or birthweight is missing altogether. Gestational age is included, but not SGA. Also, authors might consider including “frequencies” in the titles of Tables 1-4 or adding “n(%)” to table headers versus in footnotes.

  • Table 2: In naming Table 2, authors might consider language similar to Table 1.

  • Lines 214-215: The figures in Table 3 appear to be column versus row percentages; thus, the statement should read that approximately 50% of mothers with SGA had insufficient weight gain.

  • Lines 215-218: Colum percentages again: Among women with SGA, 34% were anemic, 61% were not. Among normal weight infants, 10% had anemic moms and 90% did not. Authors might want to check for this issue throughout the Results section.

  • Table 3 and 4: Both maternal and perinatal do not seem necessary in the title of these tables. To minimize confusion, authors might consider using fetal growth rather than birthweight in Table 3 title and table header.

  • Line 251: Earlier tables show that gestational age at birth, prenatal care, and mode of delivery were associated with both outcomes, but these are not in the regression models. See prior comment on Lines 168-172.

  • Lines 255-258: Chronological age was associated with LBW, but not with SGA. Authors might want to recap all of the significant findings from the regression, i.e., hemoglobin, weight gain, suburban/rural residence.

  • Lines 260-261: This seems redundant with previous sentence so appears unnecessary.

  • Line 262: Authors report odds ratios in Table 5, so they could refer throughout to greater odds versus risk, when discussing findings.

  • Lines 263-264. Authors might include a possible explanation for why OR for chronological age in prediction of LBW is greater than the OR for gynecological age.

  • Line 276: It would be helpful if authors specified the “birth outcomes” to which they are referring.

  • Lines 287-288: Finding in Table 2 seems to indicate that girls with GA<3 were on average older (12.6 years) at age of menarche than girls with GA>=3 whose average age at menarche was 11.3 yrs.

  • Lines 301-305: How do findings reported in Table 1 — statistically significant difference in age at menarche between girls <15 (11.3 years) and those 15 and older (11.7 years) — fit with this explanation?

  • Lines 306-309: Authors argue that younger age at menarche is protective against LBW and SGA — What are some of the implications of the decreasing age at menarche seen in Mexico for public health and clinical care?

  • Lines 379-380: Elaborating on how sociodemographics may make a difference would be helpful here.

  • Lines 381-383: Was hemoglobin tested at all prior to the 3rd trimester? Authors might consider the possibility that anemia may have been present earlier, but not caught until it was measured in 3rd trimester.

  • Line 395-398: Authors could add a hypothesis about the effect of place of residence as an significant independent predictor.

  • Line 425: Authors can indicate that the results apply to both LBW and SGA.

  • Lines 436-437: The authors did not test SGA as a predictor of LBW or vice versa.

  • Lines 443-445: It seems these two sentences can be combined into one, as gynecological age is an indicator of biological immaturity.

  • Lines 455-458: Authors could add major finding here -- Consider young gynecological age, not just young chronological age, as a risk factor for adverse outcomes.

Author Response

Reviewer 1

Review Report

Biomedicines-3704218:

Low birth weight or small for gestational age in neonates of adolescent mothers: Effect of gynecological and chronological age

We would like to express our sincere gratitude to Reviewer 1 for their thorough and valuable review, which has significantly contributed to enriching our manuscript. Additionally, we deeply appreciate the time and effort dedicated to conducting this review. Below, we present detailed responses to each of the raised comments.

General Impressions:

Adolescents’ increased risk for adverse pregnancy outcomes is a major public health concern, particularly in lower resourced countries where adolescent pregnancy remains relatively high. Examining a spectrum of biological and socio-ecological factors that increase risk can provide important insights for promoting healthier reproductive outcomes in adolescents. The main objective of this study is to examine the relation of chronological age and gynecological age with low birth weight (LBW) and small-for-gestational-age (SGA) in a sample of adolescent mothers, adjusting for biological and socio-ecological risk factors. The main finding that gynecological age is independently associated with LBW and SGA emphasizes the importance of considering biological immaturity, not just chronological age, in studies of adolescent pregnancy and in public health and health care efforts to promote healthier outcomes. Overall, the manuscript is well written, the research methods seem appropriate, the literature cited is mostly within 5-10 years of publication, and the conclusions follow from the results. This study could make an important contribution to the literature in this area. Improving clarity in the Methods will enhance the manuscript for publication.

Specific Comments:

Abstract

Background: Authors could highlight the importance of considering biological maturity (i.e., gynecological age) in addition to chronological age in adolescent pregnancy.

Response: the importance of considering biological maturity was highlighted in the summary.

Objective: Authors should include SGA in the study objective.

Response: the term PEG was included in the study objective.

Methods: Authors might specify that the study uses a retrospective cohort design, and that low birth weight was assessed using Intergrowth standards. As a retrospective study, they also might want to edit the statement that they “followed” 1242 adolescents.

Response: it was specified in the summary section that a retrospective cohort design was used and that low birth weight was evaluated using Intergrowth standards.

There seems to discrepancies between the list of control variables in the Abstract and those noted in Table 5 where the regression results are reported.

Response: thank you for the observation, it was verified that there were no discrepancies between the list of control variables in the abstract and in Table 5.

Results: Authors might check whether the journal requires all of the regression-related statistics to be reported in the Abstract.

Response: the journal's guidelines suggest presenting the main results that support the objective of the study.

Given their study objective, authors also could point out that chronological age was not a significant predictor of LBW

Response: it was noted that chronological age was not a significant predictor of small for gestational age.

Conclusion: Authors might place the menarche statement after the initial conclusion about biological immaturity, since it is part of the results.

Response: the proposed suggestion is accepted.

The importance of considering sociodemographic factors in prenatal care is a bit confusing, because these were used primarily as control variables, rather than being the focus of study.

Response: the conclusions were rewritten in order to eliminate confusion.

Keywords: SGA is missing. It is unclear why pathogenesis is included.

Response: the keyword pathogenesis was removed and the phrase small for gestational age was included.

Introduction

Lines 77-78: A definition of gynecological age would be quite helpful to readers.

Response: the definition of gynecological age was included in the introduction section.

Lines 83-84: Authors could clarify results of Gibbs et al, i.e., which factor – low GA or young maternal age – was predictive of LBW? How was the result linked to biological immaturity in their study?

Response: the results of the study by Gibbs et al. that relate young maternal age and biological immaturity with low birth weight were clarified.

Lines 85-86: It is unclear what the debate is between biological and socioenvironmental factors and why that would be mentioned here, given that the study’s focus is stated in the next line: relations of gynecological and chronological age to LBW and SGA.

Answer: The text has been corrected, eliminating the assertion that refers to the debate between biological and socio-environmental factors.

Materials and Methods

Lines 91-99: The manuscript would benefit from more details about the study design and the sampling procedures. Were all data collected through record reviews, for example, or a combination of records and interviews or surveys with the participants? How were participants recruited and what non-random methods were used to create the sample? Authors might consider moving the “INPer” statement from the end of the section to the beginning.

Answer: Thank you very much for the suggestion. The procedure for recruiting study participants, as well as for data collection, has been detailed.

Lines 100-155: Procedures and Data Collection section only includes information on measurement of study variables. Including details about the study procedures would enhance the manuscript. Perhaps, this can be addressed in Lines 91-99. Defining acronyms prior to use also would be helpful to readers.

Answer: The procedures for measuring the different study variables were detailed.

Answer: The different acronyms were defined before their use.

Line 135: Gestational age traditionally is measured based on last menstrual period and date of delivery. Authors might explain why they assessed gestational age based on time of last prenatal care visit.

Answer: The error regarding the assessment of gestational age has been corrected.

Lines 142-144: Table 3 indicates hemoglobin was assessed in 3rd trimester; this could be clarified here where the measure is initially described. Lower hemoglobin in 3rd trimester may not be indicative of hemoglobin levels throughout the pregnancy course.

Answer: The reason for evaluating the average hemoglobin concentration at 24 weeks of gestation was specified in the text. Furthermore, it was mentioned that the hemoglobin concentration determined between the end of the second and the beginning of the third trimester is not indicative of hemoglobin concentrations throughout the entire course of the pregnancy.

Lines 146-147: Definition of gestational age offered here is inconsistent with earlier definition in Line 135.

Answer: The error in the definition of gestational age has been corrected

Lines 150-151: Gestational age is described a third way here.

Answer: The error was corrected.

Lines 168-172: Authors ensure that the list of independent and control variables mentioned here match those listed in the model in Table 5. It also is unclear how variables were selected for the regression models. Based on the bivariate analyses, there were other variables significantly associated with the outcomes (e.g., mode of delivery), which were not included in the models.

Reply: Thank you for the question. We decided not to include some variables, such as type of delivery, as they may be an outcome of birth weight rather than an exposure variable. We selected variables that, during or before pregnancy, may affect birth weight.

Readers may find Figure 1 somewhat confusing. It is labeled as the distribution of chronological age and gynecological age, but it depicts age at menarche and chronological age. The overlap of the distributions between ages 12-15 also presents some challenges in understanding the figure. Authors might consider whether this figure is necessary.

Answer: Figure 1 was removed.

Lines 184-185: The two outcome variables are missing from Table 1 (and Table 2). LBW or birthweight is missing altogether. Gestational age is included, but not SGA. Also, authors might consider including “frequencies” in the titles of Tables 1-4 or adding “n (%)” to table headers versus in footnotes.

Answer: Thank you for the suggestion; the proposed observations were implemented.

Table 2: In naming Table 2, authors might consider language similar to Table 1.

Answer: The change suggested by the reviewer was made.

Lines 214-215: The figures in Table 3 appear to be column versus row percentages; thus, the statement should read that approximately 50% of mothers with SGA had insufficient weight gain.

Answer: The statement was corrected.

Lines 215-218: Colum percentages again: Among women with SGA, 34% were anemic, 61% were not. Among normal weight infants, 10% had anemic moms and 90% did not. Authors might want to check for this issue throughout the Results section.

Answer: The problem observed by the reviewer was verified and corrected in the results section.

Table 3 and 4: Both maternal and perinatal do not seem necessary in the title of these tables. To minimize confusion, authors might consider using fetal growth rather than birthweight in Table 3 title and table header.

Answer: The proposal was accepted, and the change was made.

Line 251: Earlier tables show that gestational age at birth, prenatal care, and mode of delivery were associated with both outcomes, but these are not in the regression models. See prior comment on Lines 168-172.

Answer: Gestational age at birth, prenatal care, and mode of delivery were included in the regression models.

Lines 255-258: Chronological age was associated with LBW, but not with SGA. Authors might want to recap all of the significant findings from the regression, i.e., hemoglobin, weight gain, suburban/rural residence.

Answer: All significant findings from the regression models in the manuscript were recapped.

Lines 260-261: This seems redundant with previous sentence so appears unnecessary.

Answer: The unnecessary sentence was removed.

Line 262: Authors report odds ratios in Table 5, so they could refer throughout to greater odds versus risk, when discussing findings.

Answer: Thank you for the observation; the term 'risk' was removed throughout the text.

Lines 263-264. Authors might include a possible explanation for why OR for chronological age in prediction of LBW is greater than the OR for gynecological age.

Answer: A possible explanation was included as to why the odds ratio for chronological age in predicting low birth weight is higher than the odds ratio for gynecological age.

Line 276: It would be helpful if authors specified the “birth outcomes” to which they are referring.

Answer: The birth outcomes to which the sentence refers were specified.

Lines 287-288: Finding in Table 2 seems to indicate that girls with GA<3 were on average older (12.6 years) at age of menarche than girls with GA>=3 whose average age at menarche was 11.3 yrs.

Answer: The age of menarche in both groups is corroborated and is correct.

Lines 301-305: How do findings reported in Table 1 — statistically significant difference in age at menarche between girls <15 (11.3 years) and those 15 and older (11.7 years) — fit with this explanation?

Answer: A probable explanation is provided for the significant difference in the average age of menarche in both groups and the results obtained.

Lines 306-309: Authors argue that younger age at menarche is protective against LBW and SGA — what are some of the implications of the decreasing age at menarche seen in Mexico for public health and clinical care.

Answer: An explanation is proposed as to under what conditions a young age at menarche is protective against LBW (Low Birth Weight) and SGA (Small for Gestational Age).

Answer: The implications of the decrease in age at menarche for public health and clinical care are provided.

Lines 379-380: Elaborating on how sociodemographic may make a difference would be helpful here.

Answer: The text is corrected, as there is no controversy in the scientific literature. Rather, the trend points to the demonstration that both the biological immaturity of the adolescent and sociodemographic factors are associated with different maternal and neonatal outcomes. In conclusion, maternal and neonatal outcomes are multifactorial.

Lines 381-383: Was hemoglobin tested at all prior to the 3rd trimester? Authors might consider the possibility that anemia may have been present earlier, but not caught until it was measured in 3rd trimester.

Answer: The explanation is provided as to why the hemoglobin concentration was analyzed on average at 24 weeks of gestation.

Line 395-398: Authors could add a hypothesis about the effect of place of residence as a significant independent predictor.

Answer: A hypothesis is provided regarding the effect of place of residence as an independent predictor.

Line 425: Authors can indicate that the results apply to both LBW and SGA.

Answer: It was indicated that the results apply to both LBW (Low Birth Weight) and SGA (Small for Gestational Age).

Lines 436-437: The authors did not test SGA as a predictor of LBW or vice versa.

Answer: We did not include both variables in the models due to their high correlation, as SGA (Small for Gestational Age) is dependent on birth weight, which could be redundant. In the Results section, we added a bivariate correlation between birth weight in grams and growth and intergrowth percentiles, with the following findings: Rho 0.736, p <0.001 according to Spearman's correlation.

Lines 443-445: It seems these two sentences can be combined into one, as gynecological age is an indicator of biological immaturity.

Answer: The two sentences were combined into one.

Lines 455-458: Authors could add major finding here -- Consider young gynecological age, not just young chronological age, as a risk factor for adverse outcomes.

Answer: The finding that young gynecological age and young chronological age are risk factors for adverse outcomes was added.

Reviewer 2 Report

Comments and Suggestions for Authors
  1. Main research question

The research question was explicitly stated in the objective (lines 28-30): "To evaluate the relationship between chronological age and gynecological age with birth weight in newborns of Mexican adolescent mothers." The paper effectively sought to determine whether biological immaturity (a short interval between menarche and conception) was a more significant risk factor for adverse birth outcomes (LBW and SGA) than a mother's age in years. This question was central to the manuscript and was addressed throughout.

  1. Originality and relevance

The manuscript's originality lies in its direct comparison of gynecological and chronological age as predictors for both LBW and SGA within a single, large cohort of Mexican adolescents. While the negative effects of adolescent pregnancy are well-documented, the distinction between biological and chronological age is a nuanced and important area of research that remains debated. Studies have often focused on one or the other, but few have compared them with this level of detail in a Latin American population.

This research addressed a specific gap in the literature concerning Mexico. The authors correctly stated that despite Mexico having a high rate of adolescent pregnancy, the specific impact of these different age metrics requires "further investigation" (line 28). By conducting the study at the National Institute of Perinatology (INPer), a major referral center, the authors provided valuable data from a key population.

A particularly relevant, though surprising, finding was the potentially protective effect of early menarche against LBW/SGA. This challenges the common assumption that all markers of early development in this context are negative and warrants significant attention and debate.

  1. Methodology and study design

The retrospective cohort design was appropriate for the research question. However, several aspects require improvement and clarification:

  • Sampling and generalizability (Lines 91-92, 415-418): The primary weakness is the "non-random sampling" at a single institution (INPer). The authors acknowledged that this "could limit the generalization of the findings" (lines 415-416). While they argued that INPer served a broad population, this did not overcome the potential for selection bias. Women referred to this high-specialty center may differ systematically from the general population of pregnant adolescents in Mexico, potentially having more complex cases or different socioeconomic backgrounds. The authors should elaborate more on this limitation in the Discussion section, cautioning readers more strongly against generalising the prevalence rates and even the risk ratios to all of Mexico.
  • Study period discrepancy (Lines 91, 175): There is a contradiction in the stated data collection period. Line 91 lists it as "January 2018 to December 2023," while line 175 states it as "January 2018 and December 2024." This clerical error must be corrected for consistency.
  • Definition of prenatal care (Line 116): Prenatal care initiation is categorised simply as "before or after 24 weeks of gestation." This is an overly broad and somewhat late cutoff. Quality and adequacy of prenatal care are determined by both the timing of the first visit and the total number of visits. Other research often uses a cutoff in the first trimester or quantifies the number of visits (e.g., <6 visits). This crude dichotomisation could obscure a more complex relationship between prenatal care and birth outcomes, as suggested by the non-significant findings for this variable in the adjusted models. The authors should justify this choice or, if the data are available, re-analyse using a more refined definition.
  • Socioeconomic level measurement (Line 109): The use of "AMAI's 8X7 Rule" is appropriate, as it is a standard tool for socioeconomic classification in Mexico. No further comment is needed here.
  • Recall bias for Menarche Age (Lines 418-420): The authors correctly identified recall bias for the age of menarche as a limitation but suggested it was "minimized" because the adolescents were close in age to the event. While plausible, this could be strengthened by citing literature that quantifies the reliability of recalled menarcheal age in adolescent populations.
  1. Consistency of evidence and conclusions

The conclusions are generally consistent with the evidence presented, particularly the results from the regression models in Table 5.

  • The claims that a gynecological age <3 years increases the risk of LBW by 3.8 times (ORadj 3.799) and SGA by 2.5 times (ORadj 2.462) are directly supported by the data (lines 444-446, Table 5).
  • Similarly, the conclusion that chronological age <15 years increases the risk of LBW by 5.7 times (ORadj 5.740) is also directly supported (lines 447-448, Table 5).
  • The finding that chronological age was not a risk factor for SGA is also consistent with the non-significant p-value in Table 5 (p=0.794).

However, the conclusion that "menarche before age 11 appears to be a protective factor for both LBW and SGA" (lines 450-451) is the most controversial point. While the ORs are indeed <1.00 and statistically significant (Table 5), the biological and social explanation for this requires a more thorough discussion. The authors suggested that an earlier menarche allowed more time for biological maturation before the average age of pregnancy in this cohort (15.7 years) (lines 300-302). This is a plausible hypothesis, but the discussion would benefit from exploring alternative explanations or conflicting evidence. For instance, some studies link early menarche itself to adverse health outcomes or associate it with other risk factors like low birth weight of the mother herself. The current explanation is brief and should be expanded to provide a more balanced perspective.

  1. Tables and Figures

The quality of the tables and figures is good. They are clear, well-labeled, and effectively summarise a large amount of data.

  • Figures 1 and 2 provide useful visual summaries of the age distributions and birth weight trends.
  • Tables 1-4 are comprehensive and logically present the bivariate analyses. The use of p-values to indicate significant differences between groups is appropriate.
  • Table 5 is the most crucial table, clearly presenting the final adjusted odds ratios from the logistic regression models. The variables included in the adjustment are clearly footnoted.
  1. Caveats, weaknesses, and specific comments

In addition to the points raised above, the following should be considered:

  • Title wording: The title "Low birth weight or small for gestational age..." is slightly ambiguous. A clearer title might be "The Differential Effects of Gynecological and Chronological Age on Low Birth Weight and Small for Gestational Age..." to better reflect the comparative nature of the study.
  • Biological immaturity vs. social factors (Lines 402-410): The Discussion section rightly acknowledged the complex interplay between biological immaturity and adverse socioeconomic factors. It correctly stated that the etiology is "multifactorial." However, the study's design cannot fully disentangle these effects. Since the population was from a national referral center, it might be skewed towards higher biological risk, potentially amplifying the observed effects of age and minimising the role of socioeconomic factors that are more prevalent in the general population. This should be discussed as a possible interpretation of the findings.
  • Insufficient GWG (Line 45): The abstract stated insufficient gestational weight gain was a factor for SGA (p=0.031), but in Table 5, the CI for the OR for insufficient GWG on LBW crosses 1 (0.902-2.956) with p=0.105. The conclusion rightly focused on the SGA finding, but the authors should ensure clarity throughout the manuscript about which outcome insufficient GWG was significantly associated with after adjustment.

Author Response

Reviewer 2

Review Report

We would like to express our sincere gratitude to Reviewer 2 for their thorough and valuable review, which has significantly contributed to enriching our manuscript. Additionally, we deeply appreciate the time and effort dedicated to conducting this review. Below, we present detailed responses to each of the raised comments.

Main research question

The research question was explicitly stated in the objective (lines 28-30): "To evaluate the relationship between chronological age and gynecological age with birth weight in newborns of Mexican adolescent mothers." The paper effectively sought to determine whether biological immaturity (a short interval between menarche and conception) was a more significant risk factor for adverse birth outcomes (LBW and SGA) than a mother's age in years. This question was central to the manuscript and was addressed throughout.

Originality and relevance

The manuscript's originality lies in its direct comparison of gynecological and chronological age as predictors for both LBW and SGA within a single, large cohort of Mexican adolescents. While the negative effects of adolescent pregnancy are well documented, the distinction between biological and chronological age is a nuanced and important area of research that remains debated. Studies have often focused on one or the other, but few have compared them with this level of detail in a Latin American population.

This research addressed a specific gap in the literature concerning Mexico. The authors correctly stated that despite Mexico having a high rate of adolescent pregnancy, the specific impact of these different age metrics requires "further investigation" (line 28). By conducting the study at the National Institute of Perinatology (INPer), a major referral center, the authors provided valuable data from a key population.

A particularly relevant, though surprising, finding was the potentially protective effect of early menarche against LBW/SGA. This challenges the common assumption that all markers of early development in this context are negative and warrants significant attention and debate.

Methodology and study design

The retrospective cohort design was appropriate for the research question. However, several aspects require improvement and clarification:

Sampling and generalizability (Lines 91-92, 415-418): The primary weakness is the "non-random sampling" at a single institution (INPer). The authors acknowledged that this "could limit the generalization of the findings" (lines 415-416). While they argued that INPer served a broad population, this did not overcome the potential for selection bias. Women referred to this high-specialty center may differ systematically from the general population of pregnant adolescents in Mexico, potentially having more complex cases or different socioeconomic backgrounds. The authors should elaborate more on this limitation in the Discussion section, cautioning readers more strongly against generalising the prevalence rates and even the risk ratios to all of Mexico.

Answer: The limitations were elaborated on in the discussion section, warning readers about the limitation for generalizing our results.

Study period discrepancy (Lines 91, 175): There is a contradiction in the stated data collection period. Line 91 lists it as "January 2018 to December 2023," while line 175 states it as "January 2018 and December 2024." This clerical error must be corrected for consistency.

Answer: The error in the data collection period was corrected.

Definition of prenatal care (Line 116): Prenatal care initiation is categorised simply as "before or after 24 weeks of gestation." This is an overly broad and somewhat late cutoff. Quality and adequacy of prenatal care are determined by both the timing of the first visit and the total number of visits. Other research often uses a cutoff in the first trimester or quantifies the number of visits (e.g., <6 visits). This crude dichotomisation could obscure a more complex relationship between prenatal care and birth outcomes, as suggested by the non-significant findings for this variable in the adjusted models. The authors should justify this choice or, if the data are available, re-analyse using a more refined definition.

Answer: The suggestion is accepted, we used the World Health Organization's definition, and the analysis was performed again. The WHO definition is shown in the methods section, and the results are shown in the different tables of the results section.

Socioeconomic level measurement (Line 109): The use of "AMAI's 8X7 Rule" is appropriate, as it is a standard tool for socioeconomic classification in Mexico. No further comment is needed here.

Recall bias for Menarche Age (Lines 418-420): The authors correctly identified recall bias for the age of menarche as a limitation but suggested it was "minimized" because the adolescents were close in age to the event. While plausible, this could be strengthened by citing literature that quantifies the reliability of recalled menarcheal age in adolescent populations.

Answer: The assertion that adolescents had a more accurate recollection of their age at menarche was strengthened, as this was close to the time of the event. This is reinforced by scientific literature on the subject.

Consistency of evidence and conclusions

The conclusions are generally consistent with the evidence presented, particularly the results from the regression models in Table 5.

The claims that a gynecological age <3 years increases the risk of LBW by 3.8 times (ORadj 3.799) and SGA by 2.5 times (ORadj 2.462) are directly supported by the data (lines 444-446, Table 5).

Similarly, the conclusion that chronological age <15 years increases the risk of LBW by 5.7 times (ORadj 5.740) is also directly supported (lines 447-448, Table 5).

The finding that chronological age was not a risk factor for SGA is also consistent with the non-significant p-value in Table 5 (p=0.794).

However, the conclusion that "menarche before age 11 appears to be a protective factor for both LBW and SGA" (lines 450-451) is the most controversial point. While the ORs are indeed <1.00 and statistically significant (Table 5), the biological and social explanation for this requires a more thorough discussion. The authors suggested that an earlier menarche allowed more time for biological maturation before the average age of pregnancy in this cohort (15.7 years) (lines 300-302). This is a plausible hypothesis, but the discussion would benefit from exploring alternative explanations or conflicting evidence. For instance, some studies link early menarche itself to adverse health outcomes or associate it with other risk factors like low birth weight of the mother herself. The current explanation is brief and should be expanded to provide a more balanced perspective.

Answer: The explanation of why menarche before the age of 11 appears to be a protective factor is expanded, thus providing a more balanced perspective on the topic.

Tables and Figures

The quality of the tables and figures is good. They are clear, well labeled, and effectively summarise a large amount of data.

Figures 1 and 2 provide useful visual summaries of the age distributions and birth weight trends.

Tables 1-4 are comprehensive and logically present the bivariate analyses. The use of p-values to indicate significant differences between groups is appropriate.

Table 5 is the most crucial table, clearly presenting the final adjusted odds ratios from the logistic regression models. The variables included in the adjustment are clearly footnoted.

Caveats, weaknesses, and specific comments

In addition to the points raised above, the following should be considered:

Title wording: The title "Low birth weight or small for gestational age..." is slightly ambiguous. A clearer title might be "The Differential Effects of Gynecological and Chronological Age on Low Birth Weight and Small for Gestational Age..." to better reflect the comparative nature of the study.

Answer: The suggestion is accepted, and the manuscript title is changed to better reflect the comparative nature of the study.

Biological immaturity vs. social factors (Lines 402-410): The Discussion section rightly acknowledged the complex interplay between biological immaturity and adverse socioeconomic factors. It correctly stated that the etiology is "multifactorial." However, the study's design cannot fully disentangle these effects. Since the population was from a national referral center, it might be skewed towards higher biological risk, potentially amplifying the observed effects of age and minimising the role of socioeconomic factors that are more prevalent in the general population. This should be discussed as a possible interpretation of the findings.

Answer: The possibility was discussed that the study sample was biased towards a higher risk, which could amplify the observed effects of age and minimize the role of socioeconomic factors in the association of low birth weight or having a newborn small for gestational age.

Insufficient GWG (Line 45): The abstract stated insufficient gestational weight gain was a factor for SGA (p=0.031), but in Table 5, the CI for the OR for insufficient GWG on LBW crosses 1 (0.902-2.956) with p=0.105. The conclusion rightly focused on the SGA finding, but the authors should ensure clarity throughout the manuscript about which outcome insufficient GWG was significantly associated with after adjustment.

Answer: Clarity was ensured throughout the manuscript that insufficient gestational weight gain was only associated with having a newborn small for gestational age.

Round 2

Reviewer 2 Report

Comments and Suggestions for Authors

Glad with changes